# Partnered Excited-State Intermolecular Proton Transfer Fluorescence (P-ESIPT) Signaling for Nitrate Sensing and High-Resolution Cell-Imaging

**DOI:** 10.3390/molecules27165164

**Published:** 2022-08-13

**Authors:** Pan Ma, Fuchun Gong, Hanming Zhu, You Qian, Lingzhi He, Jiaoyun Xia, Zhong Cao

**Affiliations:** College of Chemistry and Chemical Engineering, Changsha University of Science and Technology, Changsha 410114, China

**Keywords:** 6-pyrenyl-2-pyridinone, ESIPT fluorophore, partnered-proton transfer, nitrite sensing, cell-imaging

## Abstract

Nitrite (NO_2_^−^) is a common pollutant and is widely present in the environment and in human bodies. The development of a rapid and accurate method for NO_2_^−^ detection is always a very important task. Herein, we synthesized a partnered excited-state intermolecular proton transfer (ESIPT) fluorophore using the “multi-component one pot” method, and used this as a probe (ESIPT-F) for sensing NO_2_^−^. ESIPT-F exhibited bimodal emission in different solvents because of the solvent-mediated ESIPT reaction. The addition of NO_2_^−^ caused an obvious change in colors and tautomeric fluorescence due to the graft of NO_2_^−^ into the ESIPT-F molecules. From this basis, highly sensitive and selective analysis of NO_2_^−^ was developed using tautomeric emission signaling, achieving sensitive detection of NO_2_^−^ in the concentration range of 0~45 mM with a detection limit of 12.5 nM. More importantly, ESIPT-F showed the ability to anchor proteins and resulted in a recognition-driven “on-off” ESIPT process, enabling it to become a powerful tool for fluorescence imaging of proteins or protein-based subcellular organelles. MTT experimental results revealed that ESIPT-F is low cytotoxic and has good membrane permeability to cells. Thus, ESIPT-F was further employed to image the tunneling nanotube in vitro HEC-1A cells, displaying high-resolution performance.

## 1. Introduction

Nitrite (NO_2_^−^), the most common nitrogen-containing inorganic compound in nature, widely exists in the environment and human bodies [1,2]. Although NO_2_^−^ is a very important signal molecule and plays important roles in normal biological processes such as blood flow regulation, hypoxic nitric oxide homeostasis and intrauterine growth restriction [3,4], however, excessive NO_2_^−^ is highly toxic to human bodies [5]. For instance, NO_2_^−^ can result in normal blood oxygen-carrying hemoglobin insufficiently oxidizing methemoglobin, thus tissue hypoxia is caused by losing the oxygen-carrying capacity [6,7]. NO_2_^−^ is a carcinogen, and the mechanism of carcinogenicity suggests that NO_2_^−^ and secondary amines, tertiary amines, amides and other reactions generate strong carcinogenic nitrosamines in the gastric acid environment [8]. Additionally, the secondary product of nitrosamines can also be transported through the placenta to the fetus and cause them teratogenic, premature delivery, growth retardation, defects, etc. [9,10]. Therefore, fast, sensitive and selective detection of NO_2_^−^ is significant for analytical application in biological conditions.

To date various analytical methods are available for the monitoring of NO_2_^−^ in aqueous solutions including chromatographic analysis [11], capillary isotachophoresis [12], cyclic voltammetry [13], chemiluminescence [14], surface-enhanced Raman spectroscopy [15] and fluorescence spectrophotometry [16]. Although these methods can achieve accurate and reliable results, they are not suitable for routine determination of NO_2_^−^ in environmental and biological samples due to some of the limitations such as high detection cost, complicated and expensive equipment required, complex and time-consuming sample pre-treatment, and high requirements for the professional level of operators.

NO_2_^−^ sensing methods based on molecular probes have attracted extensive attention due to their high sensitivity, selectivity, simple operability and low-cost advantage [17,18,19]. A great number of NO_2_^−^ sensors were developed for the detection of NO_2_^−^ based on nitrite-mediated reactions. Among these, aromatic *o*-diamines were frequently used as the reactive sites and recognition receptors in the probe designs [20]. This strategy is typically based on the diazotization of a suitable aromatic amine by acidified nitrite solution with the subsequent coupling reaction and provides a highly colored azo-chromophore, from which the NO_2_^−^ concentration can be evaluated [21,22]. From this Griess basis, Ramaiah constructed a novel probe using BODIPY as a reporter that selectively recognizes NO_2_^−^ through a distinct color change from bright blue to intense green with a detection limit of 20 ppb [23]. By coupling rhodamine fluorophore with *o*-phenylenediamine, a “turn-on”-type fluorogenic probe was reported which can monitor trace amounts of NO_2_^−^ in water as low as 4.6 ppb [24]. In addition, an alternative probe (NT555) for the Griess and DAN assays was also developed to detect NO_2_^−^, which exhibited superior detection kinetics and sensitivity [25]. At present, the reported fluorescent sensors for NO_2_^−^ rely on the analogous mechanism with the Griess assay and have to match the strong acid condition (pH < 3) [26,27] and some of them even require a low-temperature condition (0 °C) [28,29]. Consequently, these probes for NO_2_^−^ generally pose the following major challenges: (1) The *o*-diamine-based reaction sites and/or the resulting products of benzotriazole are popularly pH-sensitive, and the protonation of benzotriazole under acidic conditions may bring a false response; (2) These analyses would suffer possible interference from potentially oxidizing or reducing agents in bio-samples, such as ascorbic acid (AA) and superoxide; (3) The formation of benzotriazoles often requires a long time, meaning longer response times for a response to NO_2_^−^; (4) The preparation of probes usually involves tedious synthetic steps. More recently, many NO_2_^−^ assay-based nanostructure probes emerged [30,31,32,33]. Most of them are obtained from the nanoparticles grifted with recognized groups to detect NO_2_^−^. Some of these approaches involve complicated procedures that are expensive, toxic to cells and may suffer from the aggregation and precipitation of nanoparticles when used [34,35,36]. Therefore, it is very necessary to develop a highly selective, fast responding and easily synthesized fluorescent probe for the convenient measurement of NO_2_^−^.

Fluorescence imaging technology, with its inherent advantages of noninvasiveness, excellent selectivity, high sensitivity and real-time analysis, has become a powerful tool to understand dynamic biological processes in living cells. Despite the growing interest in the development of fluorescent probes for live-cell imaging, challenges to be tackled still remain [37,38,39,40]. As far as we know, there are no reports on the use of small organic molecules as fluorescent probes for imaging tunneling nanotubes in live cells. 

Herein, we reported a robust and convenient excited-state intermolecular proton transfer probe (ESIPT-F) for detecting nitrite and imaging cells with high-resolution performance (Figure 1). ESIPT-F was synthesized using the "four-component one-pot" method. The optical properties of ESIPT-F and its response to common anions were investigated using UV–vis and fluorescence measurements, demonstrating its specific response to NO_2_^−^. The recognition mechanism of ESIPT-F to NO_2_^−^ in the protic solvent suggests that the reaction of ESIPT-F with NO_2_^−^ suppressed the ESIPT process. Excitingly, ESIPT-F can also anchor proteins and concert ESIPT reaction, enabling it to be used as a powerful tool for protein-based tunneling nanotube imaging in vitro HEC-1A cells. The high efficiency in the preparation of ESIPT-F is rather remarkable, with a one-pot reaction, no additional catalyst is needed.

## 2. Results and Discussion

### 2.1. Absorption and Fluorescence Spectra of ESIPT-F

The spectroscopic properties of ESIPT-F and its response to NO_2_^−^ were investigated in a DMF-HEPES mixed system (4/1, pH 4.5) As shown in the absorption spectra (Figure 2a), ESIPT-F exhibited two obvious absorption bands at 282 nm and 378 nm, respectively, which may be assigned to the π–π* transition and the long conjugation present in the free ESIPT-F. Upon the addition of NO_2_^−^, the absorption peaks at 282 nm and 378 nm decreased appreciably and the color of the ESIPT-F solution converts from brown to slightly yellowish (inset in Figure 2a). The UV–vis spectral results indubitably suggest the reaction of ESIPT-F with NO_2_^−^. The decrease in absorption could be attributed to the suppression of the ESIPT process due to the reaction of ESIPT-F with NO_2_^−^.

The fluorescence performance of ESIPT-F in different solvents and its fluorescence change in the presence of NO_2_^−^ were also investigated. As shown in Figure 2c, ESIPT-F displayed blue fluorescence at 453 nm in aprotic solvents including DMF, THF, ethyl acetate and acetonitrile; when exposed to the protic solvents such as water, methanol, ethanol and acetic acid, a long-wavelength green emission at 497 nm can be observed (Φ = 0.645). This unique spectra performance may suggest that ESIPT-F undergoes tautomerization between the enol and keto forms resulting from a solvent-mediated ESIPT reaction, which is similar to that of the reported compound [40]. When the emission wavelength was fixed at 497 nm, two excitation peaks testified to the existence of both the enol and keto forms observed at *λ*_max_ 285 and 385 nm in the excitation spectrum of the ESIPT-F (Figure 2a). The peak at 385 nm could be assigned to the tautomeric form of ESIPT-F, which originated from the ESIPT reaction in the excited state. Upon addition of the different NO_2_^−^ concentrations, the tautomeric fluorescence of ESIPT-F was remarkably suppressed with a blue shift from 497 nm to 455 nm (Figure 2b). Simultaneously, the emission change of the solutions can be easily visualized by the naked eye under a UV lamp (inset in Figure 2b). The obvious "on-off" type switch may be attributed to the reaction of NO_2_^−^ with ESIPT-F, which resulted in the blocking of the ESIPT process and C=N/OH isomerization. Figure 2d displays the change of emission spectra of ESIPT-F in DMF-HEPES solutions at different pHs upon excitation at 280 nm. The fluorescence is centered at *λ*_em_ = 497 nm. As the pH increases, the intensity of the band slightly changes. This shows that in the range of 3-9, pH has little effect on the tautomeric fluorescence of ESIPT-F.

### 2.2. Electrochemical Response of ESIPT-P to NO_2_^−^

A large number of redox reactions are frequently accompanied by concerted proton transfer (proton-coupled electron transfer, PCET). In our case, the ESIPT reaction of the probe involved in the proton transfer may be relevant to the electron transfer. As shown in Figure 3, the free ESIPT-F possesses a clear cathode peak at −0.07 V vs. SCE, corresponding with the reduction peak of the H^+^-transfer of the ESIPT-F. Upon the addition of NO_2_^−^, the reduction bands decreased obviously, implying that NO_2_^−^ may react with the amino group of ESIPT-F and suppress the H^+^-transfer and PCET process. These results demonstrated that ESIPT-F furnished a selective recognition to NO_2_^−^.

### 2.3. Theoretical Evaluation of the Recognition of ESIPT-F to Proteins

Inspired by the protic solvent-mediated ESIPT case, we assume that some molecules with the hydrogen receptor and donor system (D-A) can be recognized by ESIPT-F via hydrogen binding. Thus, we studied the binding properties of ESIPT-F to proteins using FBS as a model protein. As shown in Figure 4, ESIPT-F can image not only HAC-1A cells but also the FBS in the culture medium. As a control, no fluorescence emission was observed in the field of view stained with 4′,6-diamidino-2-phenylindole (DAPI). This result demonstrated that ESIPT-F can specifically identify proteins.

To explore the ESIPT emission mechanism of the fluorescent probe, we pictured the frontier molecular orbitals (FMOs) theory by means of (TD) DFT calculations. As shown in Figure 5a, for both the tautomers of ESIPT-F, HOMO and LUMO are the π and π* characters, respectively. 

The HOMO of the enol tautomer has a larger electronic projection over the O_d_ atom. However, in its LUMO picture, there is no electronic projection over the O_d_ atom. The HOMO picture of the keto tautomer also exhibits a higher electronic density projection over the O_d_ atom. Interestingly, electronic projection over the N*_a_* atom of the lactam tautomer increases in the LUMO picture, which thus theoretically indicates a favorable ESIPT phenomenon, but not a GSIPT reaction. In addition, a molecular docking simulation was also carried out in virtue of the docking software with the method of AutoDock vina. The BSA from the PDB database (ID: 4f5s) was selected as the research model. As shown in Figure 5b, both the enol and keto conformers of ESIPT-F can attach to the polypeptide and tend to insert into different pockets. The enol form ESIPT-F allowed an anchor in the specific cavity of BSA by hydrogen-binding to the amino acid residues of Lys-76, while the keto conformers tend to attach on the Cys-75. 

In addition, the enol and keto forms of ESIPT-F can bind to a BSA molecule together. These modeling studies suggested that probes may be favorably inserted into a pocket near the end of a polypeptide chain. The results of the molecular docking simulation are consistent with the protein staining experiment, demonstrating that ESIPT-F possesses a high affinity to BSA.

Based on the spectroscopic and electrochemical measurements and theoretical evaluation, we speculate that the sensing mechanism of ESIPT-F was because of the suppression of excited-state intermolecular proton transfer (ESIPT). The proposed mechanism between the probe and guest NO_2_^−^ is illustrated in Figure 1. NO_2_^−^ could react with the keto conformers of ESIPT-F and suppress the ESIPT process, resulting in the decrease in tautomeric fluorescence at 497 nm. When the concentration of NO_2_^−^ is high, the ESIPT reaction was seriously inhibited. Therefore, a new peak appeared at 452 nm, corresponding to normal emissions. In live cells, some biomolecules, such as proteins and nucleic acids, can mediate the ESIPT reaction, enabling ESIPT-F to exhibit tautomeric fluorescence and image protein-based subcellular organelles.

### 2.4. Selectivity Assays for Nitrite Detection

To investigate the selectivity, the probe ESIPT-F (15.0 μM) was treated with various anions (NO_2_^−^, NO_3_^−^, HSO_4_^−^, SO_4_^2−^, SO_3_^2−^, HSO_3_^−^, S_2_O_3_^2−^, S^2−^,CO_3_^2−^, HCO_3_^−^, F^−^, Cl^−^, Br^−^, I^−^) and oxidants (ClO^−^, Cr_2_O_7_^2−^, H_2_O_2_), respectively, in the phosphoric acid solution (pH = 4.5) at 25 °C after 30 min (Figure 6b). To our delight, none of the analytes used except nitrite caused obvious fluorescence quenching. At the same time, the results of anti-interference experiments showed that the high specificity of ESIPT-F to NO_2_^−^ ions was not affected by other ions or oxidizers (Figure 6a). In other words, ESIPT-F exhibits good anti-interference ability. The high sensitivity and good selectivity of the probe ESIPT-F imply the potential application for fast quantitatively measurement of NO_2_^−^ in food, environmental and biological materials.

### 2.5. Detection of NO_2_^−^ in Real Samples Using ESTIP-F

Considering the sensitive and specific response of ESTIP-F to NO_2_^−^, the fluorescence assays were established for the detection of NO_2_^−^ using ESTIP-F as a probe. The fluorescence titration experiments were also carried out using ESIPT-F as a probe for NO_2_^−^. As shown in Figure 7a, the tautomeric emission of ESTIP-F at 497 nm decreases gradually with the increase in NO_2_^−^ concentrations. The NO_2_^−^ concentration reached 45 μM, and the fluorescence band at 453 nm emerged, which is attributed to the saturation of the reaction of NO_2_^−^ with ESIPT-F. The fluorescence decrease in ESIPT-F is a linear correlation to NO_2_^−^ concentrations ranging from 0 to 45 μM (Figure 7b). The detection limit based on the reaction of ESTIP-F with NO_2_^−^ was evaluated as 12.5 nM (3σ). Obviously, these results confirmed that ESTIP-F has remarkably high sensitivity to NO_2_^−^.

To verify the applicability of this ESIPT probe in real samples, ESTIP-F was used to detect NO_2_^−^ in water and soil samples. The samples were prepared as described in the Experimental Section. The samples were spiked with standard NO_2_^−^ at different concentration levels and then analyzed using the ESTIP-F probe-based method. The results are listed in Table 1. One can see that the recovery studies of the spiked NO_2_^−^ samples determined using the proposed probe showed satisfactory results (98.6–103%) with a reasonable RSD ranging from 1.68 to 3.02%. The proposed method seems useful for monitoring NO_2_^−^ in real samples.

### 2.6. Cellular Imaging

Subsequently, we explored the application of ESIPT-F to subcellular organelle imaging in living cells. The cell imaging experiments were carried out using a Leica (Leica TCSSP5 model) scanning microscope. After removal of the medium and washing with phosphate-buffered saline (PBS, pH = 6.0), HEC-1A cells were incubated with ESIPT-F for 30 min, followed by fluorescence imaging. As shown in Figure 8a,e, in the cells stained with a 30 μM ESIPT-F containing solution, significant green fluorescence was observed. In contrast, the cells are dark and there is virtually no emission collected in 500–790 nm upon excitation at 408 nm. These results indicated that ESIPT-F was low cytotoxic and cell-membrane permeable to the tested cells. In addition, one can also see that the ingested ESIPT-F exhibited good color rendering and mainly concentrated in proteins in the cytoplasm. Interestingly, ESIPT-F can also picture the shape of the intercellular tunnel nanotube. These results established that the probe ESIPT-F is capable of imaging tunnel nanotubes, revealing its high-resolution performance.

## 3. Materials and Methods

### 3.1. Reagents and Instruments

1-Acetylpyrene, ethyl cyanoacetate, 4-fluorobenzaldehyde and ammonium acetate were purchased from Sinopharm Chemical Reagent Co., Ltd. (Shanghai, China). All chemicals were used as received from commercial sources. The stock solution of ESIPT-F (10 mM/L) was prepared by dissolving ESIPT-F in a little DMF and the volume of 100 mL was set with a HEPES buffer solution (10 mM, pH 7.2). Working solutions were obtained by diluting the corresponding stock solutions to an appropriate volume with the same HEPES buffer whenever required. The human endometrial adenocarcinoma cell line (HEC-1-A) was obtained from Shanghai Institutes for Biological Sciences (Shanghai, China).

The fluorescence experiments were carried out on an F-7000 fluorescence spectrophotometer (Hitachi, Tokyo, Japan). UV–vis absorption spectra were recorded with a Cary 60 UV–vis spectrophotometer (Agilent Technologies, Australia). NMR spectra were acquired with a Bruker AVB-400 MHz NMR spectrometer (Bruker biospin, Fällanden, Switzerland). Fourier transform infrared (FT–IR) spectra in KBr were recorded with a WQF-510 FT–IR spectrometer (Beijing Rayleigh Analytical Instrument Co., Ltd., Beijing, China). Electrochemical measurements were carried out using a potentiostat and galvanostat (Model Auto-lab, PGSTAT30, Ecochemie, Utrecht, The Netherlands) connected to a Pentium IV personal computer (with GPES software, Hanzhou, China). A conventional three-electrode system was employed with a platinum wire as the counter (or auxiliary) electrode and a saturated calomel electrode (SCE) as the reference electrode. A glassy carbon electrode was used as the working electrode (Azar Electrode Co., Tehran, Iran). All experiments were carried out at room temperature. 

### 3.2. Synthesis of ESIPT-F

(4-(4-Fluorophenyl)-2-oxo-6-(pyren-1-yl)-1,2-dihydropyridine-3-carbonitrile) was synthesized according to the reported method [41] with a minor modification. Briefly, a mixture of 1-acetylpyrene (3.7 g, 0.015 mol), p-fluorobenzaldehyde, (1.86 g, 0.015 mol), ethyl cyanoacetate (1.7 g, 0.015 mol) and ammoniumacetate (7.0 g, 0.09 mol) was heated, with stirring, in a microwave reactor at 110 °C (dynamic power 50–60W) for 30 min. The resulting yellowish-brown solid was washed with cold water, re-crystallized from hot ethanol and dried under vacuum. Yield: 66%. ^1^H NMR (400 MHz, DMSO) δ 13.20, 8.38, 8.28, 8.21, 8.19, 8.15, 8.14, 8.10, 7.96, 7.86, 7.84, 7.44, 7.42, 7.40, 6.73. ^13^C NMR (101 MHz, DMSO) δ 165.01, 162.74, 162.12, 159.12, 132.83, 132.55, 131.38, 131.29, 129.36, 128.75, 127.61, 126.66, 126.36, 125.05, 124.38, 124.15, 123.97, 116.46, 116.24, 109.91. ESI-MS(*m*/*z*) calcd. for C_28_H_15_FN_2_O [M-1] 414.43; found[M-1] 414.12 (see Appendix A).

### 3.3. Analytical Procedure

An appropriate amount of ESIPT-F stock solution (20 μL) was diluted into 5 mL with a HEPES buffer solution (10 mM, pH 4.5) in a 10 mL graduated tube. A certain amount of NO_2_^−^ standard solutions (or sample solutions) was then added to the test tubes. After setting to 10 mL with the same HEPES buffer, the resulting mixture was shaken thoroughly to react for 10 min at room temperature prior to the fluorescence measurements. Simultaneously, the controls without NO_2_^−^ standard solutions or sample solutions were acquired from corresponding solvents. Finally, the fluorescence intensity of the tautomeric emission peaks (F_497_) for the test solution and the reagent blank (F_497_) were directly recorded with an F-7000 fluorescence spectrophotometer with an excitation wavelength of 365 nm.

### 3.4. Preparation of Real Samples

The tap water and soil were collected from our campus. The wastewater and river water were obtained from the waterway on our campus and Xiangjiang River (Changsha, China), respectively. For the sample preparation, the water samples (100 mL, three parallel samples) were filtered through a 0.45 μm membrane filter, and their pH value was adjusted to 4.0 by hydrochloric acid (1.0 M). The soil samples (100 g) were dispersed with 100 mL water and filtered through a 0.45 μm membrane filter. The resulting samples were then used for determination.

### 3.5. MTT Assays

HEC-1A cells were used for a model cell line to evaluate the in vitro cytotoxicity of ESIPT-F using MTT assay. Firstly, HEC-1A cells were incubated in a 96-well plate (5000 cells/well) which was added with a 100 μL DMEM solution containing 10% fetal bovine serum (FBS) under humidified 5% CO_2_ atmosphere at 37 ℃. After incubation (24 h), the medium was replaced and different concentrations of ESIPT-F (0, 10, 20, 40, 60, 80, 100 µM) were added for another day of incubation. Next, the 50 μL test solution containing 10 mg·mL^−1^ MTT reagents was added to each well and further incubated at 37 ℃ for an additional 4 h. Finally, the culture medium was removed and 150 μL DMSO was then added, and shaken for 15 min in the dark. The resulting mixtures were employed for UV–vis spectrometry test by a microplate reader at the wavelength of 490 nm.

### 3.6. Protein Staining and Cell-Imaging

Protein staining and the intracellular distribution of ESIPT-F against HEC-1A cells were investigated using a laser scanning confocal microscope (CLSM). Typically, HEC-1A cells were added to a 35 mm glass-bottom culture dish at a density of 1.5 × 10^4^ cells per dish, letting them grow overnight in a 2 mL DMEM solution with 10% FBS with 5% CO_2_ under humidified atmosphere at 37 °C for 36 h. After that, the original medium mass was replaced and 2 mL fresh medium containing ESIPT-F (50 μM) was then added to each dish. Followed by a 12 h incubation, ice-cold PBS buffer (20 mM, pH 7.4) was added to wash the remaining ESIPT-F. After washing with the same ice-cold PBS buffer three times, the resulting cells and the FBS in culture medium were then observed under CLSM.

## 4. Conclusions

In summary, a novel ESIPT fluorescent probe for detecting NO_2_^−^ ions was successfully designed and synthesized in this work. ESIPT-F can undergo an ESIPT reaction mediated by not only protic solvents but also proteins, exhibiting dual-model emission. The reaction of NO_2_^−^ with ESIPT-F leads to the suppression of the ESIPT process and tautomeric fluorescence quenching, which is used as signaling for sensing NO_2_^−^. More importantly, ESIPT-F can anchor proteins and concert the ESIPT process, making it a powerful tool for protein and subcellular organelle imaging. Thus, this ESIPT fluorescent probe would provide a simple, rapid and convenient method to monitor NO_2_^−^ ions. In addition, ESIPT-F shows application prospects in fluorescence imaging of protein-based subcellular organelle imaging.

## Data Availability

Not applicable.

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
