# Peer review of "Partnered Excited-State Intermolecular Proton Transfer Fluorescence (P-ESIPT) Signaling for Nitrate Sensing and High-Resolution Cell-Imaging"

_molecules, 2022, doi:10.3390/molecules27165164_

Round 1
Reviewer 1 Report
In this work, the authors developed an ESIPT mechanism-based probe and tried to show its applications in water as well as in the cellular level for nitrite sensing. The introduction part is good, which well-stated the current progress of related probes with reasonable comparison, it was suggested to add a scheme with these probes’ structure which can make the readers a more direct view of them.
Besides, the rest parts of this manuscript are not ready for publication yet.
1. the skeleton of the paper needs to be adjusted. Scheme 1 should move to the beginning of the paper. Without knowing the probe’s structure, it will be confusing to see how it works first.
2. characterization, Fig s1, and s2 have not matched the requirement of NMR data, no peaks’ position, no integration.
3. the authors should have a standard when managing or editing their figures, same font style, font size, and font color. Also, both the X-axis and Y-axis of the figures should be taken care of when conducting magnification or minification.
Other errors:
1. Figure 1c, line and indicator color different (blue and light blue).
2. caption of Figure 1, font size.
3. line 144, Mm should be Mm.
4. line 146, no space between number and unit.
5. figure 3f, control means?
6. In figure 4a, there are no details of the orbital calculation data, like exactly how much ev of HOMO and LUMO? CI value? Oscillator strength?
7. figure 4b, if the author wants to discuss the binding environment, then this data should be presented as a stick model.
8. figure 5 used EtOH-HEPES instead of DMF-HEPES, any special reason for that?
9. line 226, 0 to 45 M, this is an unacceptable mistake.
10. line 251, the “data not shown” actually need to be presented.
11. line 258, ESIPT-P8?
Please also check for similar errors else in the paper as well.
-end
Author Response
Dear Prof.,
We are truly grateful to you for your comments which would be valuable for the improvement of our manuscript. According to your comments, we have revised the manuscript in order to meet the requirements of Molecules. All the changes in the manuscript were clearly marked in red. The comments made by the reviewers were explained point-to-point and some important issues were addressed as following.
General comments: In this work, the authors developed an ESIPT mechanism-based probe and tried to show its applications in water as well as in the cellular level for nitrite sensing. The introduction part is good, which well-stated the current progress of related probes with reasonable comparison, it was suggested to add a scheme with these probes’ structure which can make the readers a more direct view of them. Besides, the rest parts of this manuscript are not ready for publication yet.
Response: Thank you very much for your comments. We have revised our manuscript according to your suggestions in order to meet the requirements of Molecules. The explanations and treatments of the questions raised by you are as follows.
Specific comments and response
- the skeleton of the paper needs to be adjusted. Scheme 1 should move to the beginning of the paper. Without knowing the probe’s structure, it will be confusing to see how it works first.
Response: According to your suggestion, Scheme 1 was placed at the beginning of the paper in the revised manuscript.
- characterization, Fig s1, and s2 have not matched the requirement of NMR data, no peaks’ position, no integration.
Response: We added the clearer pictures in Fig.S1 and S2 in the revised manuscript.
- the authors should have a standard when managing or editing their figures, same font style, font size, and font color. Also, both the X-axis and Y-axis of the figures should be taken care of when conducting magnification or minification.
Response: Thank you for your suggestion. We have tried to scale the Figures according to a standard in the revised manuscript.
Other errors:
- Figure 1c, line and indicator color different (blue and light blue).
We corrected the color of the indicating line in Figure 1c in the revised manuscript.
- caption of Figure 1, font size.
We corrected the caption of Figure 1, font size in the revised manuscript.
- line 144, Mm should be Mm.
The “Mm” was corrected to “Mm.” in the caption of Fig. 2 in the revised manuscript.
- line 146, no space between number and unit.
We added space between the number and unit in line 146 in the revised manuscript.
- figure 3f, control means?
The caption of figure 3f was corrected to “dark fields”
- In figure 4a, there are no details of the orbital calculation data, like exactly how much ev of HOMO and LUMO? CI value? Oscillator strength?
We added orbital energy of HOMO and LUMO in figure 4a in the revised manuscript.
- figure 4b, if the author wants to discuss the binding environment, then this data should be presented as a stick model.
We attempt to discuss the binding affinity of ESIPT-F to proteins which is related to cell-imaging.
- figure 5 used EtOH-HEPES instead of DMF-HEPES, any special reason for that?
The writing errors “EtOH-HEPES” were corrected to “DMF-HEPES” in the revised manuscript.
- line 226, 0 to 45 M, this is an unacceptable mistake.
The writing error “45 M” was corrected to “45 mM” in the revised manuscript.
- line 251, the “data not shown” actually need to be presented.
The statement “data not shown” was deleted in the revised manuscript.
- line 258, ESIPT-P8?
The writing error “ESIPT-P8” was corrected to “ESIPT-F”.
Finally, we would like to express our sincere thanks to you for giving us a chance to revise the manuscript and their valuable comments and suggestions to improve the quality of this paper. If you have any question, please contact with me without any hesitation.
Thank you very much for your time and consideration.
Sincerely yours,
Dr. Gong Fu-Chun
Reviewer 2 Report
Dear Editor
Authors reported a robust and convenient excited-state intermolecular proton transfer probe (ESIPT-F) for detecting nitrite and imaging cells with high-resolution performance. ESIPT-F was synthesized using "four-component one-pot" method. The optical properties of ESIPT-F and its response to common anions were investigated using UV-vis and fluorescence measurements, demonstrating its specific response to NO2- . The recognition mechanism of ESIPT-F to NO2 - in the protic solvent is suggested that the reaction of ESIPT-F with NO2 - suppressed the ESIPT process. It was a powerful tool for protein-based tunneling nanotube imaging in vitro HEC-1A cells. The. paper can be accepted after some additions and corrections.
1- Uniform styles should be used for Figures and their titles (For example fig 6).
2- Response time experiments should be given.
3- pH studies were performed.
4- Some references should be added to fluorescent sensors, Energy transfer, cell imaging and applications.
https://doi.org/10.1016/j.jphotochem.2021.113456
https://ieeexplore.ieee.org/abstract/document/8573901
https://pubs.rsc.org/en/content/articlelanding/2015/ra/c4ra12874e/unauth
https://doi.org/10.1016/j.jphotochem.2021.113655
https://link.springer.com/article/10.1007/s10895-019-02456-3
Best Regards
Author Response
Dear Prof.,
We are truly grateful to you for your comments which would be valuable for the improvement of our manuscript. According to your comments, we have revised the manuscript in order to meet the requirements of Molecules. All the changes in the manuscript were clearly marked in red. The comments made by the reviewers were explained point-to-point and some important issues were addressed as following.
General comments: Authors reported a robust and convenient excited-state intermolecular proton transfer probe (ESIPT-F) for detecting nitrite and imaging cells with high-resolution performance. ESIPT-F was synthesized using "four-component one-pot" method. The optical properties of ESIPT-F and its response to common anions were investigated using UV-vis and fluorescence measurements, demonstrating its specific response to NO2- . The recognition mechanism of ESIPT-F to NO2 - in the protic solvent is suggested that the reaction of ESIPT-F with NO2 - suppressed the ESIPT process. It was a powerful tool for protein-based tunneling nanotube imaging in vitro HEC-1A cells. The paper can be accepted after some additions and corrections.
Response: Thank you very much for your comments. We have revised our manuscript according to your suggestions in order to meet the requirements of Molecules. The explanations and treatments of the questions raised by you are as follows:
Specific comments and response
- Uniform styles should be used for Figures and their titles (For example fig 6).
Response: According to your suggestion, we have tried our best to use a uniform style for Figures in the revised manuscript.
- Response time experiments should be given.
Response: Response time experiments were performed the data is not shown in paper.
- pH studies were performed.
Response: The pH studies have carried out and the data was given in Fig.1d.
- Some references should be added to fluorescent sensors, Energy transfer, cell imaging and applications.
Response: According to your suggestion, we added two references in the revised manuscript.
([39] Z. Dikmen, O.Turhan, M. Yaman, V. Bütün, V. An effective fluorescent optical sensor: Thiazolo-thiazole based dye exhibiting anion/cation sensitivities and acidochromism. J. Photochem . Photobio. A: Chem. 419, (2021) 113456; [40] K. Ahmed Nuri, O. Mustafa, G. Ersin, A Novel Fluorescent Chemosensor for cu (II) Ion: Click Synthesis of Dual-Bodipy Including the Triazole Groups and Bioimaging of Yeast Cells. J .Fluor. 29 (2019):1321-1329.)
Finally, we would like to express our sincere thanks to you for giving us a chance to revise the manuscript and their valuable comments and suggestions to improve the quality of this paper. If you have any question, please contact with me without any hesitation.
Thank you very much for your time and consideration.
Sincerely yours,
Dr. Gong Fu-Chun
Round 2
Reviewer 1 Report
The work has improved compared to the last version. A few more issues need to be addressed before publishing.
1. I have suggested adding a figure of existed probe’s structure in the introduction section (just insert one table or figure in the SI), however, the author just choose to ignore my words instead to state anything.
2. the NMR data need to involve the peak position and integration in the picture, which the author also just simply ignore my comment. (or they have uploaded the wrong SI file).
3. For the molecular orbital calculation, I have commented that the CI value, Oscillator strength were necessary to show the molecule’s photophysical property, which can be easily found in the Gaussian program used. Which is ignored by the authors again in this version.
4. Even I pointed out the unit spelling issue. Still, some mM are presenting as Mm.
-End
Author Response
Dear Prof.,
We are truly grateful to you for your comments which would be valuable for the improvement of our manuscript. According to your comments, we have revised the manuscript in order to meet the requirements of Molecules. All the changes in the manuscript were clearly marked in red. The comments made by the reviewers were explained point-to-point and some important issues were addressed as following.
General comments: The work has improved compared to the last version. A few more issues need to be addressed before publishing.
Response: Thank you very much for your comments. We have revised our manuscript according to your suggestions in order to meet the requirements of Molecules. The explanations and treatments of the questions raised by you are as follows.
Specific comments and response
- I have suggested adding a figure of existed probe’s structure in the introduction section (just insert one table or figure in the SI), however, the author just choose to ignore my words instead to state anything.
Response: According to your suggestion, we added a figure (Fig.1) of existed probe’s structure in the introduction section in the revised manuscript.
- the NMR data need to involve the peak position and integration in the picture, which the author also just simply ignore my comment. (or they have uploaded the wrong SI file).
Response: We added the peak position and integration in the picture of Fig.S1 in the revised manuscript.
- For the molecular orbital calculation, I have commented that the CI value, Oscillator strengthwere necessary to show the molecule’s photophysical property, which can be easily found in the Gaussian program used. Which is ignored by the authors again in this version.
Response: Thank you for your suggestion. We added the CI value, Oscillator strength in Fig. 5 in the revised manuscript.
- Even I pointed out the unit spelling issue. Still, some mMare presenting as Mm.
Response: We corrected the writing error “Mm” in the revised manuscript.
Finally, we would like to express our sincere thanks to you for giving us a chance to revise the manuscript and their valuable comments and suggestions to improve the quality of this paper. If you have any question, please contact with me without any hesitation.
Thank you very much for your time and consideration.
Sincerely yours,
Dr. Gong Fu-Chun
Reviewer 2 Report
Can be accepeted
Author Response
thank you for the comments